# Electricity Production and Population Dynamics of Microbial Community in a Co-Culture of Iron Mine Soil Biofilm and *Shewanella oneidensis* MR-1 with Anode as Electron Acceptor

**DOI:** 10.3390/microorganisms13102383

**Published:** 2025-10-16

**Authors:** Huimei Chi, Jiayi Bai, Man Feng

**Affiliations:** 1State Key Laboratory of Digital Medical Engineering, School of Biological Science and Medical Engineering, Southeast University, Nanjing 211189, China; 2Department of Bioengineering, Imperial College London, South Kensington Campus, London SW7 2AZ, UK; 3State Key Laboratory of Palaeontology and Stratigraphy, Nanjing Institute of Geology & Palaeontology, Nanjing 210008, China

**Keywords:** iron mine soil biofilm, Shewanella oneidensis MR-1, co-culture, microbial fuel cell, population dynamics

## Abstract

Microbial communities that develop within biofilms on electrodes are necessary for the proper functioning of the microbial electrochemical system. However, the mechanism through which an exogenous exoelectrogen influences the population dynamics and electrochemical performance of biofilms remains unclear. In this study, we explored the community structure dynamics and electrochemical characteristics of iron mine soil biofilm co-cultured with Shewanella oneidensis MR-1, with the anode as the electron acceptor, and compared the results with those of iron mine soil biofilms alone on the anode. Shewanella oneidensis MR-1 improved the electrochemical activity of microbial biofilms, resulting in a higher maximum power density of 195 ± 8 mW/m^2^ compared with that of iron mine soil (175 ± 7 mW/m^2^) and *Shewanella* (88 ± 8 mW/m^2^) biofilms individually. The co-cultured biofilms could perform near the highest power density for a longer duration than the iron mine soil biofilms could. High-throughput 16S rRNA gene sequencing of the biofilms on the anode indicated that the relative abundance of Pelobacteraceae in the co-culture system was significantly (*p* = 0.02) increased, while that of Rhodocyclaceae was significantly (*p* = 0.008) decreased, compared with that in iron mine soil biofilms. After continuing the experiment for two months, the presence of *Shewanella oneidensis* MR-1 changed the predominant bacteria of the microbial community in the biofilms, and the relative abundance of *Shewanella* was significantly (*p* = 0.02) decreased to a level similar to that in iron mine soil. These results demonstrate that Shewanella oneidensis MR-1 could improve the performance of iron mine soil biofilms in electrochemical systems by altering the composition of the functional microbial communities.

## 1. Introduction

Electrochemically active bacteria can transfer electrons from respiration directly to extracellular insoluble electrodes and minerals through the extracellular electron transfer (EET) process [1,2,3]. The first extracellular respiratory process, depending on EET, to be discovered was iron and manganese respiration in Shewanella and *Geobacter* species [4,5]. Multihaem c-type cytochromes involved in the dissimilatory reduction of solid metal oxides in *S. oneidensis* MR-1 and *G. sulfurreducens* utilized the EET process [6,7]. In *S. oneidensis* MR-1, the extracellular c-type cytochromes can directly reduce solid metal oxides [3,8,9,10]. Similarly, in *Geobacter*, the electrons are hypothesized to be transferred from outer membrane c-type cytochromes to an external acceptor via the type IV pili [11,12,13,14]. The Shewanella and Geobacter species are also regarded as model bacteria in bioelectrochemical systems [15].

However, pure cultures cannot produce as much electrical power in MFCs as mixed cultures do [16,17,18]. Multispecies exoelectrogenic biofilms on electrodes play a significant role in the function of bioelectrochemical systems [19]. The current generation of microbial electrochemical systems is highly dependent on the exoelectrogenic community structure. Thus, further insight into the microbial composition of biofilms on electrodes is crucial for improving the function of bioelectrochemical systems and for gaining a better understanding of extracellular electron transfer [20]. The diversity of microbial communities influences both power generation and environmental remediation [21,22,23,24,25,26]. The increase in power has been known to coincide with a significant change in the microbial community associated with the electrode [22,27]. Different anode potentials and power generation could result in additional differences in population dynamics [28,29,30]. On the contrary, it seems the operation mode does not affect the microbial community, as ascertained after conducting a microbial community analysis of the single-equivalent sediment MFC (SMFC) and the scaled-up SMFC [28]. Anyhow, an understanding of the microbial community of these systems is essential for perfecting the MFC technology [31,32].

Co-culture of nonexoelectrogen and exoelectrogen improves the electrochemical performance compared with that achieved using a pure culture [18]. Co-culturing of phototrophs and Geobacter has previously been shown to help in energy conversion from light to electricity in microbial solar cells [17]. Co-culture experiments with an electrode as electron acceptor have shown that microbial interaction can specifically convert target substrates into electricity [16,33]. Shewanella oneidensis MR-1 was supplied with a microbial fuel cell to enhance the degradation of a recalcitrant organic compound, o-xylene [34]. Similarly, cocultured systems of denitrifying anaerobic methane oxidation (DAMO) microbes and Shewanella MR-1 indicate that DAMO is involved in iron reduction with methane as a carbon source [35]. However, precisely how an exogenous electricigen influences the community structure of the exoelectrogenic biofilm and the performance of the microbial fuel cell remains to be elucidated.

In this study, we investigated the population dynamics within a co-culture of iron mine soil biofilm/Shewanella MR-1 compared with that of iron mine soil biofilm alone using Illumina HiSeq sequencing of the 16S rRNA gene. Moreover, we also considered the electrochemical performance of the co-culture system compared with that of the iron mine soil biofilm and Shewanella MR-1, when considered separately.

## 2. Materials and Methods

### 2.1. Sample Collection and Enrichment of Electroactive Biofilms from Iron Mine Soil Sample Were Collected from Meishan Iron Mine, China, 300 m Underground (Latitude: 31.53° N, Longitude: 118.43° E)

All samples were stored at −20 °C in the dark and directly transported to the laboratory by low-temperature delivery, where MFCs of iron mine soil were immediately constructed. Two-chambered electrochemical fuel cells were designed and fabricated using glass. The total working volume of one chamber was 100 mL. Carbon paper (3 cm × 3 cm) was used as an electrode after ultrasonic cleaning in acetone and deionized water, respectively. Nafion-117 (DuPont, Wilmington, DE, USA) was used as a proton exchange membrane (PEM). The anode was immersed in the anolyte (DM medium) with an iron soil. The cathode was immersed in the catholyte 50 mmol/L K_3_[Fe(CN)_6_] and dissolved in PBS buffer. A 1000-ohm resistor was connected between the anode and cathode. After running for a month at 25 °C, the carbon paper with biofilms was taken out in a super-clean bench (via UV sterilization) for the co-culture experiment.

### 2.2. Co-Culture Experiment and Electrochemical Analysis

Fifteen reactors were set up with the same method in 2.1, which were divided into three groups. Each group included five parallel MFCs. All the cathodes were carbon paper, and the catholyte was 50 mmol/l K_3_[Fe(CN)_6_], dissolved in Phosphate Buffer Solution (0.05 mol/L, pH = 7.0). The first group was operated with pure culture in the anode chamber using Shewanella MR-1 in DM medium. The second group was operated with iron soil biofilms, using carbon papers as anodes, and the anolyte was DM medium. The third co-culture group was operated with iron soil biofilms, carbon paper as anodes, and inoculated with MR-1 in DM medium. All anode chambers of MFCs were operated in anaerobic conditions by filling with nitrogen before being sterilized. A Data Acquisition Card (USB5936, Beijing Art Technology Development Co., Ltd., Beijing, China) was used to collect the data on voltage every day. The current and the power data were calculated using Ohm’s law. Cyclic voltammetry measurements of MFCs were performed during operations on a CHI600E (Chenhua, Shanghai, China) at a scan rate of 10 mV/s.

### 2.3. SEM Imaging of Biofilms on Anode

After the MFCs in 2.2 were running for two months, small pieces of carbon paper were picked in a sterile environment. The morphology of the biofilms was observed under a scanning electron microscope (SEM, Zeiss Ultra Plus, Oberkochen, Germany).

### 2.4. 16S rRNA Gene High-Throughput Sequencing

Metagenomic DNA was extracted and purified from carbon cloth anodes twice. The first sampling time was conducted one week after the MFCs in 2.2 were running, and the samples were signed as SiS1-5 (iron soil biofilms) and CoS1-5 (co-cultured biofilms). The second sampling time was two months later, after the MFCs in 2.2 were running, and the samples were signed as SiL1-5 (iron soil biofilms) and CoL1-5 (co-cultured biofilms). Metagenomic DNA was extracted and purified from carbon paper anodes by using the SunShineBioTM Microbial Genomic DNA Extraction Kit according to the manufacturer’s instructions. Using 20 ng of metagenomic DNA of the samples as template, PCR was carried out using primers 515F (5-GTGCCAGCMGCCGCGGTAA-3) and 806R (5-GGACTACVSGGGTATCTAAT-3) with the following cycling conditions: 98 °C for 3 min; 25 cycles of 98 °C for 15 s, 50 °C for 30 s, 72 °C for 30 s; 72 °C for 5 min. The final fragment of the PCR was isolated and purified by 2% agarose gel electrophoresis. Sequencing was performed on the Illumina HiSeq 2500 platform according to standard protocols (by Shanghai Personal Biotechnology Co., Ltd., Shanghai, China).

### 2.5. Analysis of 16S rRNA Gene Sequencing Data

The sequence data of 16S rRNA genes were analyzed using general methods, as described by Liu et al., 2018 [36]. The software involved included Fast Length Adjustment of SHort reads (FL-ASH) software (V1.2.7, http://ccb.jhu.edu/software/FLASH/ (accessed on 4 April 2019)) [37], Quantitative Insights Into Microbial Ecology pipeline software (Qiime version 1.9.0, http://qiime.org/ (accessed on 4 April 2019)) [38,39], and the Mothur program (http://www.mothur.org/ (accessed on 4 April 2019)) [40,41,42]. Statistical software (SPSS 17.0) was used to evaluate significant differences between iron mine soil biofilms and co-cultured biofilms. A level of *p* < 0.05 was considered to be statistically significant. The principal coordinates analysis (PCoA) plots based on the relative abundance of OTUs were generated by UniFrac algorithms. Principal components analysis (PCA) plot based on the relative abundance of OTUs was generated by the ggplots2 package of R (Version 2.15.3). The raw Illumina sequencing data were deposited in the National Center for Biotechnology Information BioProject under accession no. PRJNA530844.

## 3. Results

### 3.1. Electrochemical Activity of MFCs of Co-Cultured System

In the cyclic voltammetry curves, the iron mine soil biofilm/MR-1 co-cultures on anodes of microbial fuel cells (MFCs) exhibited the highest redox peak (Figure 1). Peak current increased in the order of iron mine soil/MR-1 co-culture, iron soil, and Shewanella MR-1. The oxidation peak was significantly higher in the co-cultured MFC than in the other MFCs. The differences among the redox peaks indicate that co-culturing of iron mine soil and MR-1 could enhance the microbial extracellular electron transfer between biofilm and anode. The maximum power density of MFCs of co-culture was found to be 195 ± 8 mW/m^2^, which was higher than that of iron mine soil (175 ± 7 mW/m^2^) and Shewanella (88 ± 8 mW/m^2^), individually. The co-culture reactor could run three cycles in 27 days in accordance with that of Shewanella. In contrast, the iron mine soil reactor ran three cycles in about 18 days, which was, by far, the shortest of the three (Figure 2). The co-cultured biofilm could perform near the highest power density for about 5 days per cycle, while the iron mine soil biofilm could only perform in the highest power density for 1 or 2 days per cycle.

### 3.2. Anode-Associated Bacterial Community Structure of the Iron Soil Biofilm and the Co-Cultured Biofilm

Upon filtering the raw tags of the samples, 51,027 to 284,267 high-quality effective tags of the 16S rRNA gene (Appendix A) were obtained for each sample, with an average length of 273 bp (Appendix A). Operational taxonomic unit (OTU) distribution of the iron mine soil and co-cultured biofilms is presented in Appendix A. The average OTUs of iron soil biofilms increased from 1355 to 1440 after running for two months. The average OTUs of co-cultured biofilms increased from 1515 to 1676 after running for two months. The average OTUs of co-cultured biofilm were less than those of the iron mine soil biofilm (Appendix A).

Table 1 shows the alpha diversity indices of the cultures, including the Simpson, Shannon, Chao1, and ACE indices. The Simpson and Shannon indices evaluate the abundance and uniformity of microbial communities [43]. After running for two months, the diversity of the bacterial community had a relatively higher density than that of the biofilm at the early stage of culturing. The Shannon indices of co-cultured biofilm were 4.89 ± 0.39 at the early stage and 5.21 ± 0.19 after running for two months. Similarly, the Shannon indices of iron soil biofilm were 6.01 ± 0.15 in the early stage and 6.55 ± 0.12 after running for two months. The diversity of the bacterial community of co-cultured biofilm had a relatively lower diversity than that of the iron soil biofilm, at both the early stage and the post-two-month stage. Furthermore, uniformity measured in the bacterial community indicated the same trend as diversity via the Simpson indices. The iron soil biofilm showed the highest uniformity among all the biofilms (see Simpson indices in Table 1). The bacterial communities of co-cultured biofilm had a relatively lower diversity (Simpson indices of 0.86–0.96) than those of the iron soil biofilms (Shannon indices of 6.01 ± 0.15 and 6.55 ± 0.12) (Table 1). Furthermore, the value of Chao1 and ACE could be used to estimate the richness of the microbial community structure [44]. The Chao1 and ACE indices of the co-cultured biofilm were lower than those of the iron soil biofilm, either in the early stage or in the two months later stage (Table 1). Thus, the diversity and uniformity had no direct relationship with electricity production in our experiment. Three clusters were indicated by a principal coordinate analysis (PCoA) according to unweighted UniFrac distance based on the relative abundance of OTUs (Figure 3). This proved a separation between the iron soil biofilm and the co-cultured biofilm, whereas there was no separation between different stages of the iron soil biofilm. The co-cultured biofilm proved to be two separate clusters. The bacterial community of the co-cultured biofilms changed substantially after running for two months. However, Co-cultivation promoted an increase in the total abundance of microbial communities, resulting in an increase in electricity production (see Figure 2 and Figure 6). 

### 3.3. Shewanella MR-1 Altered the Bacterial Structure of Iron Mine Soil Biofilms

Figure 4 shows the morphology of Shewanella MR-1 (Figure 4a), iron mine soil (Figure 4b), and co-cultured biofilms (Figure 4c). In these biofilms, *Gammaproteobacteria*, *Deltaproteobacteria*, *Clostridia*, *Alphaproteobacteria*, *Bacteroidia,* and *Betaproteobacteria* were the most predominant bacterial classes (Figure 5). Thus, the predominant class in all biofilms was Proteobacteria (Figure 5). Furthermore, the genus and family level structure of the microbial community was distinctively different between the iron soil biofilm and the co-cultured biofilm (Figure 6). In iron soil biofilms, Proteiniclasticum decreased significantly to 1.52 ± 0.52 (*p* = 0.04) after two months. Rhodocyclaceae was the most abundant bacterium, followed by Desulfuromonas, Pseudomonas, and Thermomonas. In the co-cultured biofilms, Shewanella decreased significantly to 1.47 ± 0.78 (*p* = 0.02) after two months. Pseudomonas (18.28 ± 4.85) was the most abundant bacterium in these biofilms, followed by Pelobacteraceae, Desulfuromonas, and Comamonadaceae. Desulfuromonas decreased from 11.8 ± 2.10 to 1.10 ± 0.45 after the addition of Shewanella MR-1 to the iron soil biofilm. After two months, the Desulfuromonas population recovered to 10.28 ± 4.58; however, Shewanella decreased significantly from 26.27 ± 2.83 to 1.47 ± 0.78. This result implied that Desulfuromonas and Shewanella may have a mutual exclusion effect within a certain concentration range. Pelobacteraceae was significantly higher in the co-cultured biofilm (15.98 ± 3.71 and 16.9 ± 4.69, at the early stage and post two months, respectively) than in the iron soil biofilm (2.43 ± 0.68 and 1.95 ± 0.32, respectively). Rhodocyclaceae was significantly lower in the co-cultured biofilms (0.14 ± 0.05 and 0.90 ± 0.54, respectively) than in the iron soil biofilm (9.58 ± 3.23 and 8.92 ± 2.63). This result implied that Rhodocyclaceae tended to exert a mutual exclusion effect on Shewanella MR-1 and Pelobacteraceae, while Pelobacteraceae could synergize with Shewanella MR-1. Shewanella (0.50 ± 0.06 and 2.50 ± 0.42) and Geobacter (3.68 ± 1.55 and 1.66 ± 0.59) were, however, not the most abundant genera in the iron soil biofilm.

The proportion of the top ten genera and families was higher in the co-cultured biofilms (61.79 ± 8.76 and 67.37 ± 6.50) than in the iron mine soil biofilms (45.53 ± 3.68 and 48.17 ± 2.98). Furthermore, the highest proportion (67.37 ± 6.50) was found in mature co-cultured biofilms, indicating that the proportion of predominant functional bacteria could influence the electricity generation in MFCs.

## 4. Discussion

Mixed-culture biofilm essentially produces more electricity than the pure culture catalyst [18]. In a mixed-culture biofilm community, individual species appear to play specific roles in maintaining synergistic communities [15]. In our results, Shewanella considerably improved the performance of anode-associated exoelectrogenic biofilms from iron mine soil. Not only was the power density improved, but it also extended the electricity production time (Figure 1). The structure of the functional microbial community in the iron mine soil biofilm was obviously altered following the addition of Shewanella MR-1 (Figure 2). Many factors might influence the structure of microbial communities on electrodes, including temperature, electrode modification, pH, inoculum, and ferric ion concentration [44,45,46,47,48,49,50,51,52]. Our results indicated that an exogenous exoelectrogen could also be the main factor that alters the microbial community composition on the anode. In Figure 6, after running for a period of time with the addition of Shewanella, the microbial community structure in the iron soil biofilm changed; however, the abundance of Shewanella returned to the original abundance of the soil microbial community.

Our soil sample came from an iron mine, and the microbes in it were consequently related to iron metabolism. In the iron mine soil biofilm collected from the anode, several bacterial families such as *Rhodocyclaceae*, *Pelobacteraceae*, *Desulfuromonadaceae*, and *genera* known to reduce Fe^3+^ were found [53,54]. These bacteria were all detected and showed the highest abundance near the Fe (III)/Fe (II) redox boundary [53]. Dissimilatory iron-reducing bacteria could produce electricity on an electrode through extracellular electron transfer. The typical exoelectrogens were metal-reducing bacteria such as *Geobacter* spp. and *Shewanella* spp. [4,5].

The predominant bacteria in our biofilm included Gammaproteobacteria, Deltaproteobacteria, Clostridia, Alphaproteobacteria, Bacteroidia, and Betaproteobacteria. Proteobacteria, such as Alphaproteobacteria, Betaproteobacteria, Deltaproteobacteria, and Gammaproteobacteria, were the main exoelectrogens associated with previously conducted research [15,55]. Even though several exoelectrogens proved to be useful in the MFC systems, Proteobacteria possessed the largest numbers amongst all electrochemically active bacteria [15]. In fact, Proteobacteria and Firmicutes have previously been reported to be electrochemically active, as well as to play a significant role in the EET process [56].

At the class level, there was no definite difference in community structure between the co-cultured biofilms and the iron soil biofilms. However, at the genus level, the functional exoelectrogenic communities changed significantly (Table 2). Our results demonstrate that the microbial proportions were altered considerably after the MFC had run steadily for two months (Figure 2). Furthermore, Shewanella MR-1 possibly exerts a mutual exclusion effect on Rhodocyclaceae and Desulfuromonas, while MR-1 could synergize with Pelobacteraceae. Geobacter was detected in the biofilm, but was not significant in either content or ratio (Table 2). Geobacter has previously been found to be the main exoelectrogen in the microbial communities on electrodes in some MFC systems [36,57,58,59]. The interaction between G. sulfurreducens and S. oneidensis has been reported to be likely in the presence of a hydrogen cycle, with S. oneidensis as a producer and G. sulfurreducens as a consumer [53]. G. sulfurreducens cells were observed in the planktonic phase only in the presence of S. oneidensis [53]. In our experiments, adding S. oneidensis did not seem to influence Geobacter, most probably because the latter was not the main contributor in our iron mine soil biofilms. Although the proportion of Geobacter did decrease after adding S. oneidensis, the difference was not statistically significant (Table 1). The ratio of the predominant bacteria in co-cultured biofilm was higher than that in the iron soil biofilm (Table 1), which may be implicated in the resultant improvement of the performance of MFCs. Meanwhile, the UniFrac distance based on the relative abundance of OTUs showed separation between co-culture biofilms and iron soil biofilm, which also indicated altered microbial community structure. Our results provide evidence that a single exoelectrogen is capable of modifying the exoelectrogenic microbial community. An exogenous bacterial species was discovered as one of the main factors influencing the exoelectrogenic microbial communities. Additionally, the potential interaction among the related bacterial families was revealed, which could provide further clues for designing co-culture experiments.

## 5. Conclusions

This study demonstrated that co-cultured biofilm on the anode between Shewanella and iron soil biofilm improved the performance of MFC. A maximum power density increased from 175 ± 7 mW/m^2^ to 195 ± 8 mW/m^2^, and the highest power density plateau extended to 5 days from 2 days. The oxidation peak was significantly higher in the co-cultured MFC than in the other MFCs. Shewanella MR-1 altered the microbial communities of the iron mine soil biofilms. Rhodocyclaceae was the most abundant bacterium in the iron soil biofilms, followed by Desulfuromonas, Pseudomonas, and Thermomonas. Contrastingly, Pseudomonas was the most abundant bacterium in the co-cultured biofilms, followed by Pelobacteraceae, Desulfuromonas, and Comamonadaceae. The bacterial community of the co-cultured biofilm had a relatively lower diversity than the iron soil biofilm, at both early and mature stages of culturing. The proportion of the predominant functional bacteria population in the biofilm microbial community may, thus, influence electricity generation in the MFCs. The molecular origin of the observed relationship is currently unclear and needs further study.

## Figures and Tables

**Figure 1 microorganisms-13-02383-f001:**
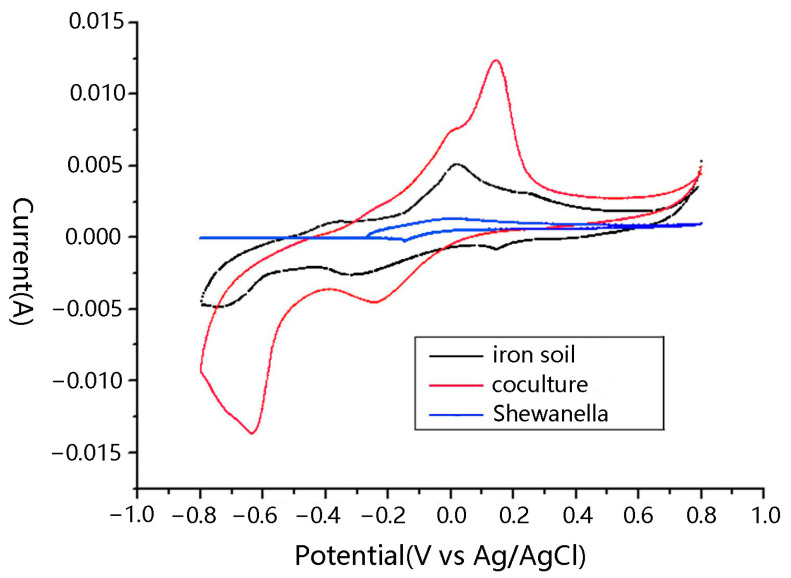
Cyclic voltammetry curves of microbial fuel cells with Shewanella MR-1, iron mine soil, and cocultured biofilms.

**Figure 2 microorganisms-13-02383-f002:**
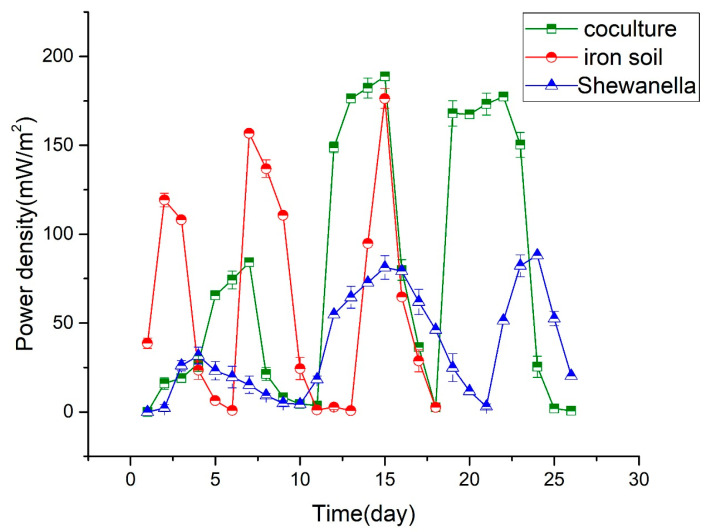
Power density curves of Shewanella MR-1, iron mine soil, and co-cultured biofilms.

**Figure 3 microorganisms-13-02383-f003:**
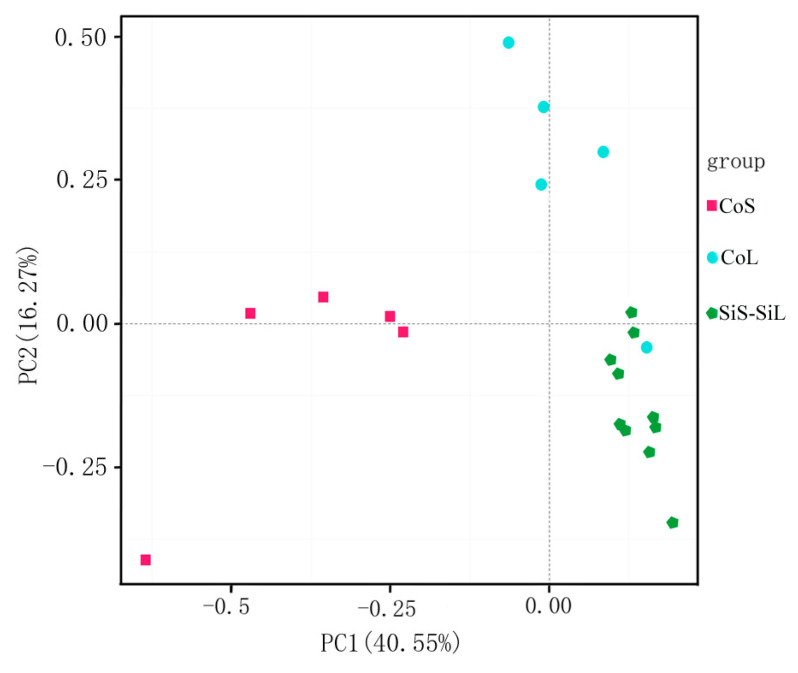
Principal coordinate analysis (PCoA) based on operational taxonomic units of iron mine soil biofilms and co-cultured biofilms in microbial fuel cells.

**Figure 4 microorganisms-13-02383-f004:**
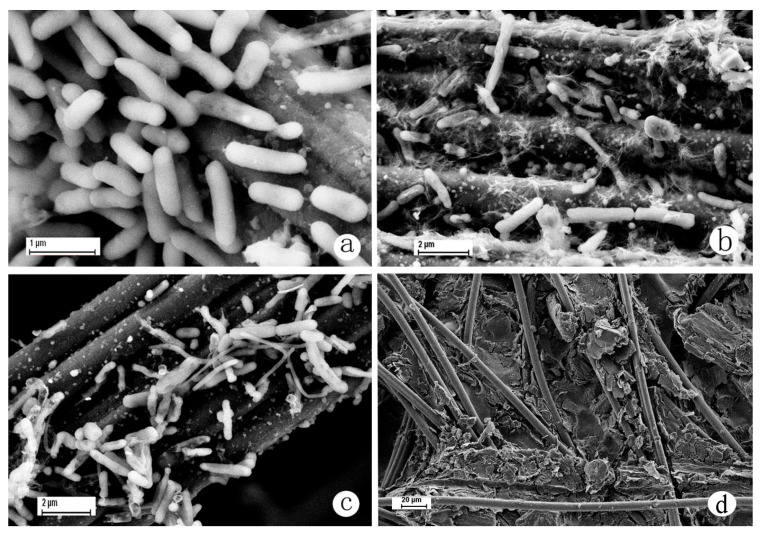
Scanning electron micrographs of biofilms of Shewanella MR-1, iron mine soil, and co-cultured biofilms on anode (**a**) Shewanella MR-1, (**b**) iron mine soil, (**c**) co-cultured of Shewanella MR-1/iron mine soil, (**d**) and empty carbon paper.

**Figure 5 microorganisms-13-02383-f005:**
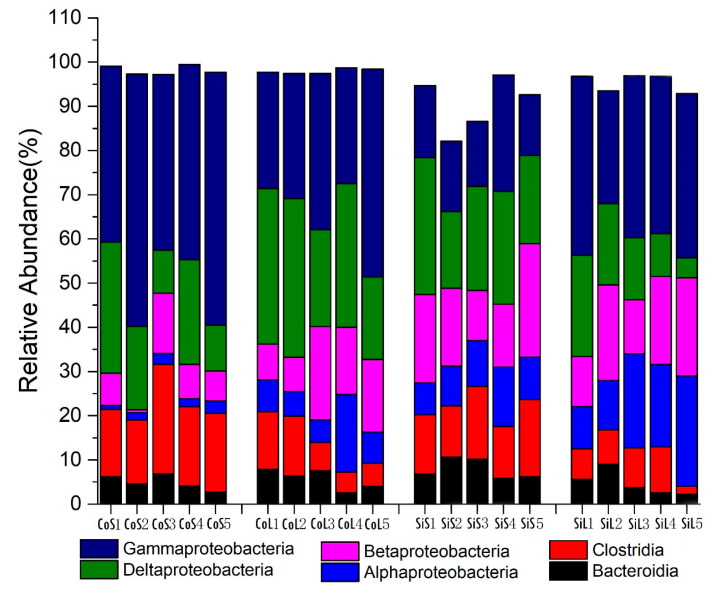
Relative abundance of 16S rRNA sequences of the iron mine soil biofilms and co-cultured biofilms in microbial fuel cells at the class level.

**Figure 6 microorganisms-13-02383-f006:**
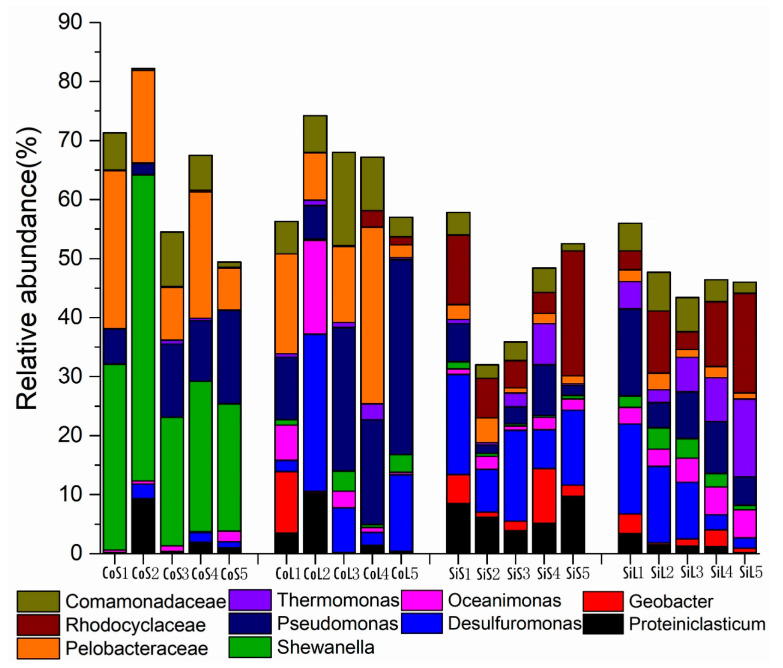
Relative abundance of 16S rRNA sequences of the iron mine soil biofilms and co-cultured biofilms in microbial fuel cells at the family and genus levels.

**Table 1 microorganisms-13-02383-t001:** Richness indices of microbial community composition in the reactors.

	Chao1	ACE	Simpson	Shannon
CoS	1698.16 ± 102.67	1658.61 ± 106.83	0.87 ± 0.04	4.89 ± 0.39
CoL	1812.95 ± 68.18	1781.50 ± 59.69	0.91 ± 0.01	5.21 ± 0.19
SiS	1879.36 ± 136.45	1859.55 ± 135.01	0.95 ± 0.00	6.01 ± 0.15
SiL	2073.69 ± 51.77	2033.65 ± 74.18	0.96 ± 0.00	6.55 ± 0.12

**Table 2 microorganisms-13-02383-t002:** Functional microbes in the biofilms on anodes.

Name of Bacteria	Level	CoS	CoL	SiS	SiL
Shewanella	Genus	26.27 ± 2.83 a	1.47 ± 0.78 b	0.50 ± 0.06 b	2.50 ± 0.42 b
Thermomonas	Genus	0.26 ± 0.13 b	1.10 ± 0.41 b	2.14 ± 1.27 b	6.66 ± 1.84 a
Geobacter	Genus	0.02 ± 0.02 a	2.14 ± 2.07 a	3.68 ± 1.55 a	1.66 ± 0.59 a
Desulfuromonas	Genus	1.10 ± 0.45 b	10.28 ± 4.58 ab	11.8 ± 2.10 a	8.46 ± 2.71 ab
Oceanimonas	Genus	0.74 ± 0.29 a	5.18 ± 2.86 a	1.56 ± 0.32 a	3.84 ± 0.42 a
Pseudomonas	Genus	9.28 ± 2.43 b	18.28 ± 4.85 a	4.22 ± 1.42 b	8.12 ± 1.88 b
Pelobacteraceae	Family	15.98 ± 3.71 a	16.9 ± 4.69 a	2.43 ± 0.68 b	1.95 ± 0.32 b
Rhodocyclaceae	Family	0.14 ± 0.05 b	0.90 ± 0.54 b	9.58 ± 3.23 a	8.92 ± 2.63 a
Proteiniclasticum	Family	2.48 ± 1.73 ab	3.14 ± 1.86 ab	6.70 ± 1.06 a	1.52 ± 0.52 b
Comamonadaceae	Family	4.52 ± 1.70 ab	7.98 ± 2.16 a	2.92 ± 0.53 b	4.54 ± 0.82 ab
Total (percent)		61.79 ± 8.76	67.37 ± 6.50	45.53 ± 3.68	48.17 ± 2.98

Note: Different letters across treatments mean a significant difference at the *p* < 0.05 level.

## Data Availability

The raw Illumina sequencing data were deposited in the National Center for Biotechnology Information BioProject under accession no. PRJNA530844.

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
