# Peer review of "Electricity Production and Population Dynamics of Microbial Community in a Co-Culture of Iron Mine Soil Biofilm and Shewanella oneidensis MR-1 with Anode as Electron Acceptor"

_microorganisms, 2025, doi:10.3390/microorganisms13102383_

Round 1

Reviewer 1 Report

Comments and Suggestions for Authors

The present paper is interesting since it examines the electricity production and population dynamics of microbial community in a co-culture of iron mine soil biofilm and She-wanella oneidensis MR-1 with anode as electron acceptor

Overall, the manuscript is well written, and the subject is interesting since authors investigated the population dynamics within a co-culture of iron mine soil biofilm/Shewanella MR-1 compared with that of iron mine soil biofilm alone, by illumina HiSeq sequencing of 16S rRNA gene.

Following are my remarks and recommendations.

Point 1: Results SECTION: Figure 3, Authors should clearly describe the redox signal observed at 0.2V and -0.2V and then give the corresponding redox equation. Moreover, the reference electrode used must be indicated in the abscise axe. Ex: Potential (V)/ref

Point 2: Authors should provide the design of the experimental set up of the MFC.

Point 3: references should be updated, and more recent ones should be added.

Point 4: Abstract SECTION should be improved. It would provide the main obtained results.

Point 5: In FIGURE 4, authors should add the SEM image of the empty carbon paper as comparison.  

Author Response

Point 1: Results SECTION:Figure 3, Authors should clearly describe the redox signal observed at 0.2V and -0.2V and then give the corresponding redox equation. Moreover, the reference electrode used must be indicated in the abscise axe. Ex: Potential (V)/ref

Response 1: We could not get complete redox signal at 0.2V and -0.2V. We have added the reference electrode in Fig.1.(Fig.3 is PCA analysis, independent of voltage).

Point 2: Authors should provide the design of the experimental set up of the MFC.

Response 2: We have provided the design of the experimental set up of the MFC in graphical abstract.

Point 3: references should be updated, and more recent ones should be added.

Response 3: We have updated some of the references highlighted in the manuscript.

Point 4: Abstract SECTION should be improved. It would provide the main obtained results.

Response 4: We have provided the main obtained results in Abstract SECTION.

Point 5: In FIGURE 4, authors should add the SEM image of the empty carbon paper as comparison.

Response 5: We have added the SEM image of the empty carbon paper as comparison in Figure.4.

Reviewer 2 Report

Comments and Suggestions for Authors

This is an interesting study of the impact of augmentation by an electrogenic microorganism on the properties an microbial community structure of a microbial fuel cell.  It is interesting that the community with the greatest output is the natural biofilm augmented with the electrogenic Shewanella strain and that there are consistent changes in the microbial community that correlate with the addition of the Shewanella strain.  I have a number of comments for the Authors’ consideration.

Major comments:

  1. It would be very interesting if the Authors were to comment on the likely changes in the abundance of the added Shewanella strain in the co-culture experiments. Whilst the genus-level analysis of the 16S rRNA gene sequence data do not show this unambiguously, the data presented in Figure 6 are consistent with a substantial decline in the relative abundance of the added strain between the two time points.
  2. Related to the above, it would be interesting if the Authors commented on how the current density achieved by the different microbial communities changes between the time two points at which the samples were taken for 16S rRNA gene analysis.

Minor comments:

  1. In the abstract and elsewhere, please give exact P values.
  2. Line 64. “shown” would be better than “exhibited”.
  3. Lines 79-80. “iron soil Iron mine soil”.  Something needs deleting here.
  4. Line 80. Please give information (e.g. GPS coordinates) that would allow the sampling site to be found on a map, or on the ground, by readers.
  5. Line 90. “super clean bench”.  More details are needed of this.
  6. Line 94. Please specify the constituents and pH of the PBS buffer.
  7. End of line 144 and several other places in the manuscript. Please take care to show superscripts correctly.
  8. Figure 1. Just one cyclic voltammogram is shown for each type of sample, although repeats were set up and analysed by other methods.  Were the cyclic voltammograms done on all samples or just one of each?
  9. Lines 145-146. Please state the number of (biological) repeats used to get the mean and standard deviations of the maximum power density values.
  10. Figure 2 and associated text. A clearer description is needed of the origin of the oscillations shown in the power density of the samples.
  11. Line 185. Add “community” before “structure”.
  12. Line 187. There is no direct relationship, but is there an inverse one? – i.e. as community diversity goes down current generation goes up?
  13. Lines 243-244. There is an agreement problem with this sentence (because “each” is singular).  More importantly, the implication that every species in every community has a specific role in synergy within the community seems unlikely.  Maybe it would be better to say something like: “individual species appear to play specific roles in maintaining synergistic communities”.
  14. Line 250. pH (lower case p).
  15. In the conclusions, it would be interesting to comment more on the functional conclusions that can be made between the community structure data and power density, even if it is only to say that the molecular origin of the observed relationship is currently unclear and needs further study.

Author Response

Major comments:

  1. It would be very interesting if the Authors were to comment on the likely changes in the abundance of the added Shewanella strain in the co-culture experiments. Whilst the genus-level analysis of the 16S rRNA gene sequence data do not show this unambiguously, the data presented in Figure 6 are consistent with a substantial decline in the relative abundance of the added strain between the two time points.

Response 1: In Fig.6, After running for a period of time with the addition of Shewanella, the microbial community structure in the iron soil biofilm changed, but the abundance of Shewanella returned to the original abundance of the soil microbial community. We have added this part in discussion.

  1. Related to the above, it would be interesting if the Authors commented on how the current density achieved by the different microbial communities changes between the time two points at which the samples were taken for 16S rRNA gene analysis.

Response 2: Not only was the power density improved, but it also extended the electricity production time (Fig. 1). The structure of the functional microbial community in the iron mine soil biofilm was obviously altered following addition of Shewanella MR-1 (Fig. 2). We have this part in discussion.

Minor comments:

  1. In the abstract and elsewhere, please give exact P values.

Response 1: We have already gave exact P values in the abstract and elsewhere.

  1. Line 64. “shown” would be better than “exhibited”.

Response 2: We have used “shown” .

  1. Lines 79-80. “iron soil Iron mine soil”.  Something needs deleting here.

Response 3: “iron soil” has been deleted.

  1. Line 80. Please give information (e.g. GPS coordinates) that would allow the sampling site to be found on a map, or on the ground, by readers.

Response 4: Latitude and Longitude has been added.

  1. Line 90. “super clean bench”.  More details are needed of this.

Response 5: We added the condition of super clean bench.

  1. Line 94. Please specify the constituents and pH of the PBS buffer.

Response 6: Constituents and pH of the PBS buffer has been specified.

  1. End of line 144 and several other places in the manuscript. Please take care to show superscripts correctly.

Response 7: We have changed show superscripts correctly.

  1. Figure 1. Just one cyclic voltammogram is shown for each type of sample, although repeats were set up and analysed by other methods.  Were the cyclic voltammograms done on all samples or just one of each?

Response 8: We have done cyclic voltammogram on all samples and then chosen the typical one of them.

  1. Lines 145-146. Please state the number of (biological) repeats used to get the mean and standard deviations of the maximum power density values.

Response 9: The number of (biological) repeats used to get the mean and standard deviations of the maximum power density values was five.

  1. Figure 2 and associated text. A clearer description is needed of the origin of the oscillations shown in the power density of the samples.

Response 10: Replace nutrient solution is the origin of the oscillations shown in the power density of the samples, which was described in material and methods part.

  1. Line 185. Add “community” before “structure”.

Response 11: We have added it.

  1. Line 187. There is no direct relationship, but is there an inverse one? – i.e. as community diversity goes down current generation goes up?

Response 12: We observed that “community diversity goes down while current generation goes up” in coculture groups. However, in iron mine biofilms we didn`t find that.

  1. Lines 243-244. There is an agreement problem with this sentence (because “each” is singular).  More importantly, the implication that every species in every community has a specific role in synergy within the community seems unlikely.  Maybe it would be better to say something like: “individual species appear to play specific roles in maintaining synergistic communities”

Response 13: We used this expression“individual species appear to play specific roles in maintaining synergistic communities”.

  1. Line 250. pH (lower case p).

Response 14: We used lower case p.

  1. In the conclusions, it would be interesting to comment more on the functional conclusions that can be made between the community structure data and power density, even if it is only to say that the molecular origin of the observed relationship is currently unclear and needs further study.

Response 15: We considered the reviewer`s suggestion and added it.